Report

# The transcription factor PAX5 activates human LINE1 retrotransposons to induce cellular senescence

Huanyin Tang [ID][1,4], Jiaqing Yang [ID][1,4], Junhao Xu[1], Weina Zhang[1], Anke Geng [ID][1], Ying Jiang[2] & Zhiyong Mao [ID][1,3][✉]

## Abstract

As a hallmark of senescent cells, the derepression of Long Interspersed Elements 1 (LINE1) transcription results in accumulated LINE1 cDNA, which triggers the secretion of the senescence-associated secretory phenotype (SASP) and paracrine senescence in a cGAS-STING pathway-dependent manner. However, transcription factors that govern senescence-associated LINE1 reactivation remain ill-defined. Here, we predict several transcription factors that bind to human LINE1 elements to regulate their transcription by analyzing the conserved binding motifs in the 5′-untranslated regions (UTR) of the commonly upregulated LINE1 elements in different types of senescent cells. Further analysis reveals that PAX5 directly binds to LINE1 5′-UTR and the binding is enhanced in senescent cells. The enrichment of PAX5 at the 5′-UTR promotes cellular senescence and SASP by activating LINE1. We also demonstrate that the longevity gene SIRT6 suppresses PAX5 transcription by directly binding to the PAX5 promoter, and overexpressing PAX5 abrogates the suppressive effect of SIRT6 on stress-dependent cellular senescence. Our work suggests that PAX5 could serve as a potential target for drug development aiming to suppress LINE1 activation and treat senescence-associated diseases.

**Keywords** Cellular Senescence; LINE1; PAX5; SIRT6
**Subject Category** Chromatin, Transcription & Genomics

## Introduction

Cellular senescence is a state characterized by irreversible cell cycle arrest during which cells lose their original functions. This process is accompanied by a reorganization of chromatin, specifically in the heterochromatin regions, leading to looser and more accessible heterochromatin structures. Such regions often encompass a substantial amount of long interspersed nuclear element 1 (LINE1), which accounts for ~17% of the human genome (Treangen and Salzberg, 2012). In healthy cells, the histone lysine methyltransferase SUV39H1 binds to the LINE1 region, promoting the formation of H3K9me3 (Bulut-Karslioglu et al, 2014), while the RB1-EZH2 complex facilitates the establishment of H3K27me3 in the same region (Ishak et al, 2016). These processes collectively foster the formation of heterochromatin to suppress LINE1 transcription. Contrarily, senescent cells or cells in aging tissues often exhibit derepression of LINE1 (Cecco et al, 2013a, 2013b). Prior research indicates that, in senescent cells, RB1 expression is significantly downregulated, whereas the DNA exonuclease TREX1, which degrades LINE1 cDNA, is also downregulated. These shifts, coupled with the upregulation of the transcription factor FOXA1 activating LINE1 transcription, subsequently trigger the IFN-1 response and foster the secretion of the senescence-associated secretory phenotype (SASP) in a cGAS-STING pathway-dependent manner (Cecco et al, 2019; Thomas et al, 2017; Simon et al, 2019). In addition, RNA transcribed from LINE1 DNA impedes the activity of SUV39H1, leading to heterochromatin erosion and elevated SASP expression (Valle et al, 2022). Previous studies have laid the groundwork for understanding how LINE1 retrotransposons can trigger SASP. SASP may induce senescence in an autocrine fashion, reinforcing senescence within the originating cell (Tasdemir and Lowe, 2013; Campisi, 2013). In addition, these factors are involved in inducing senescence in neighboring cells, which is called "paracrine senescence". Furthermore, SASP factors can also participate in juxtacrine signaling transduction by directly interacting with adjacent cells to propagate senescence, thereby amplifying the cellular senescence signaling cascade (Acosta et al, 2013; Hoare et al, 2016).

Considering the adverse effects resulting from the derepression of LINE1 as previously discussed, great efforts have been made to identify factors that are implicated in the inhibition of LINE1 to attenuate the aging process. For instance, a previous study indicates that the longevity gene SIRT6 binds to the 5'-UTR region of LINE1, leading to the mono-ADP ribosylation of KAP1. This promotes the binding of KAP1 with HP1α and consequently stimulates the formation of heterochromatin in the LINE1-enriched region (Meter et al, 2014). Similarly, as a scaffold protein, another longevity gene SIRT7, represses LINE1 transcription to delay stem cell aging by

[1]Shanghai Key Laboratory of Maternal Fetal Medicine, Clinical and Translational Research Center of Shanghai First Maternity and Infant Hospital, Frontier Science Center for Stem Cell Research, School of Life Sciences and Technology, Tongji University, Shanghai 200092, China. [2]Shanghai Key Laboratory of Signaling and Disease Research, School of Life Sciences and Technology, Tongji University, Shanghai 200092, China. [3]Present address: School of Life Sciences and Technology, Tongji University, Shanghai 200092, China. [4]These authors contributed equally: Huanyin Tang, Jiaqing Yang. ✉E-mail: zhiyong_mao@tongji.edu.cn

interacting with nuclear lamina proteins and heterochromatin proteins to repress the LINE1-enriched region (Bi et al, 2020). Furthermore, SIRT7 facilitates the interaction between the LINE1-enriched region and the nuclear lamina by deacetylating H3K18, thus preserving the transcriptionally repressed state (Vazquez et al, 2019). In addition to SIRT6 and SIRT7, BMAL1 also exhibits a similar scaffolding function to SIRT7 in mitigating LINE1 transcription (Liang et al, 2022).

Although a large number of transcription factors—such as YY1 (Becker et al, 1993), RUNX3 (Yang et al, 2003), p53 (Wylie et al, 2016), SRY (Tchénio et al, 2000), MeCP2 (Muotri et al, 2010), OCT4, NANOG (Kunarso et al, 2010), SOX2 (Kuwabara et al, 2009), CTCF, CEBPA, and CEBPB (Sun et al, 2018)—have been discovered to regulate LINE1 transcription in different biological contexts, in senescent cells, FOXA1 is the sole factor known to bind to LINE1 and boost LINE1 transcription. Nevertheless, given the intricate and diverse mechanisms underlying various cellular senescence processes, along with the absence of a single universal marker for all senescent cells, it suggests that FOXA1 may not be the exclusive transcription factor activating LINE1 in all senescent cells. Thus, it is plausible that additional transcription factors may be involved in the regulation of LINE1 transcription during cellular senescence (Cecco et al, 2019). This assumption is also supported by the tissue specificity of transcription factor expression. Hence, uncovering additional transcription factors implicated in the regulation of LINE1 in senescent cells could further advance our understanding of the mechanisms and tissue specificity underlying cellular senescence. Simultaneously, it could offer potential drug targets for the therapeutic intervention of tissue-specific diseases related to cellular senescence.

The transcription factor PAX5 belongs to the paired box (PAX) family and is predominantly expressed in the B-lymphoid lineage. PAX5 plays a critical role in B-cell development (Nutt et al, 1999). Throughout the maturation of B-lymphocytes, PAX5 is instrumental in activating the transcription of B lineage-specific genes, while concurrently repressing the expression of lineage-promiscuous genes (Cobaleda et al, 2007). Both deregulation and genetic alterations of PAX5 have been implicated in the onset of leukemogenesis (Gu et al, 2019; Shah et al, 2013). However, paradoxically, PAX5 can also function as a tumor suppressor in other types of cancers such as hepatocellular carcinoma, in which it modulates p53 transcription by binding to its promoter (Liu et al, 2011). Similarly, in esophageal squamous cell cancer, its participation in regulating p53 signaling pathway enhances chemosensitivity (Zhang et al, 2022). Despite these findings, the relationship between PAX5, cellular senescence, and LINE1 transcription activation remains sparsely investigated.

In this study, we predicted the transcription factors enriched in the LINE1 transcripts that are upregulated across different types of senescent cells. By utilizing a LINE1 5'-UTR luciferase fusion reporter, we screened the candidate transcription factors, and identified PAX5 as a novel positive regulator of LINE1 promoter activity. In senescent cells, we observed an increase in PAX5 expression, along with enhanced recruitment of PAX5 to the 5'-UTR regions of LINE1 elements to promote LINE1 transcription, thereby reinforcing cellular senescence and promoting SASP expression. We also demonstrated that SIRT6 was present at the PAX5 promoter and suppressed PAX5 expression to repress LINE1 transcription, thereby inhibiting cellular senescence.

# Results and discussion

## A number of transcription factors are predicted to bind to upregulated LINE1 region in different types of senescent cells

To identify transcription factors that regulate LINE1 in the context of cellular senescence, we first utilized SQuIRE to quantify the expression of different LINE1 subfamilies under conditions of stress-induced premature senescence (SIPS), replicative senescence (RS), and oncogene-induced senescence (OIS) using RNA-Seq data (Casella et al, 2019; Yang et al, 2019; Data ref: Casella et al, 2019). There are two replicates in each type of cellular senescence. In agreement with previous reports (Colombo et al, 2018; Cecco et al, 2019; Vazquez et al, 2019; Bi et al, 2020), most LINE1 subfamilies exhibited increased expression in senescent cells (Fig. 1A). Next, we performed locus-level quantification of LINE1 transcript expression and utilized DESeq2 for differential expression analysis. This analysis identified LINE1 variants that were significantly upregulated in each type of senescent cell. We observed overlap among the differentially upregulated LINE1 variants across different types of cellular senescence, suggesting these may potentially influence cellular senescence (Fig. 1B). To predict potential transcription factors that regulate the expression of these LINE1 variants, we selected those that were differentially upregulated in at least two types of cellular senescence. We then utilized the sequence from −200 bp to 1000 bp of the LINE1 sequence as input. Using STREME, we predicted transcription factors that are potentially enriched in these sequences (Fig. 1C). Among these factors, YY1, and FOXA1 have been previously reported as regulators of LINE1 transcription, validating the reliability of our prediction approach, while ZNF460, SP3, KLF4, FOSL2, and PAX5 are novel factors that potentially regulate LINE1 transcription (Fig. 1C). Subsequently, using the sequence analysis tool SEA, we analyzed the distribution of transcription factor binding motifs on these differentially upregulated LINE1 variants. The PAX5 binding motif was enriched around the transcription start site (TSS) of the LINE1 sequence. Meanwhile, the binding motif of ZNF460 was enriched at both the beginning and end of the LINE1 5'-UTR, and the SP3 binding motif was predominantly found in the middle of the LINE1 5'-UTR. However, no significant enrichment of FOSL2 was observed in the LINE1 start region (Fig. 1D). In addition, we examined the frequency of transcription factor motif occurrences on sequence with at least one match within the upregulated LINE1 variants associated with senescence. The occurrence of a single match was the most common. Compared to motifs of FOSL2 and ZNF460, those of SP3, KLF4, YY1, PAX5, and FOXA1 were likely to appear once. Furthermore, the motifs of PAX5 and SP3 exhibited multiple matches across multiple LINE1 variants (Fig. 1E).

## PAX5 directly regulates LINE1 transcription

In an effort to verify the roles of the five identified novel transcription factors in LINE1 transcription, we utilized a previously reported LINE1 5'-UTR and luciferase fusion reporter vector to measure the impact of the transcription factor candidates on LINE1 5'-UTR transcriptional activity (Fig. 2A) (Athanikar et al, 2004). Surprisingly, overexpression of PAX5 significantly enhanced LINE1 5'-UTR transcriptional activity even when compared to

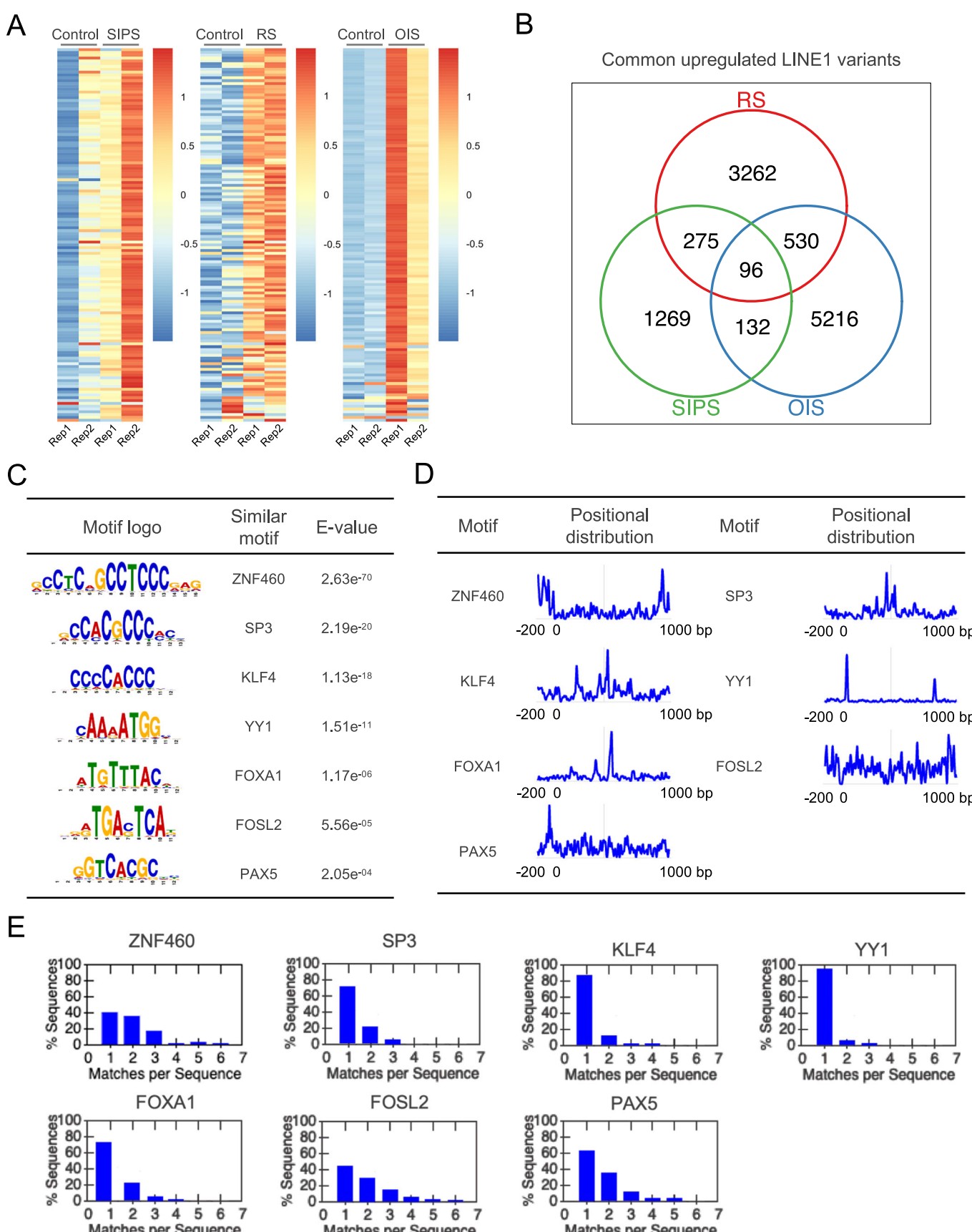

**Figure 1.  Prediction of transcription factor binding motifs enriched in commonly upregulated LINE1 variants in senescence.**

(A) Heatmaps showing the expression of LINE1 subfamilies in cells in the state of stress-induced premature senescence (SIPS), replicative senescence (RS), and oncogene-induced senescence (OIS). (B) A Venn diagram showing commonly upregulated LINE1 variants in SIPS, RS, and OIS. (C) List of enriched candidate transcription factor binding motif identified from commonly upregulated LINE1 variants. (D) The positional distribution of the best matches to the motif in the LINE1 5′-UTR. (E) The distribution of the number of matches to the motif in the LINE1 5′-UTR.

FOXA1, a previously reported potent senescence-associated LINE1 activator (Fig. 2B,C). We subsequently used FIMO to predict PAX5 binding sites on the LINE1 5′-UTR based on its binding motif. The prediction suggested that the majority of reliable PAX5 binding sites were located within the first 300 bp region of the LINE1 5′-UTR (Fig. 2D). We also analyzed the PAX5 ChIP-Seq in GM12878 cell and found PAX5 enriched at the beginning of full-length LINE1 (Fig. 2E). These are consistent with prior results from SEA, indicating that PAX5 binding motif enrichment sites are concentrated in the first half of LINE1 elements (Fig. 1D). In addition, the potential PAX5 binding sites also overlapped with the CpG island region of LINE1 which are demethylated in senescent cells (Fig. 2D) (Penzkofer et al, 2017; Ramini et al, 2022). By using a ChIP assay, we demonstrated that endogenous PAX5 bound to the LINE1 5′-UTR directly (Fig. 2F, Appendix Fig. S1). To further validate the predicted binding sites of PAX5 to 5′-UTR, we mutated the PAX5 binding sites as predicted by FIMO and measured the change in transcriptional activity. Mutations in these PAX5 binding sites led to a substantial reduction in the transcriptional activity of LINE1 5′-UTR. Particularly, when the 244–255 bp or 250–261 bp region was mutated, the promoting effect of PAX5 on LINE1 5′-UTR transcriptional activity was diminished by more than 90% (Fig. 2G). These findings suggest that PAX5 binds to LINE1 5′-UTR and directly modulates the transcription of LINE1.

## PAX5 potentiates LINE1 transcription and promotes cellular senescence

To investigate whether PAX5 plays important roles in activating LINE1 in the context of senescence, we induced cells into the state of SIPS with 10 Gy X-ray irradiation and collected cells on the tenth day. The mRNA and protein expression of PAX5 were examined using RT-qPCR and Western blot. These results indicated a significant increase in both mRNA and protein expression of PAX5 in HCA2 SIPS cells (Fig. 3A,B). We also confirmed our finding with another widely used fibroblast cell line, IMR90-hTERT (Fig. 3C). Furthermore, by analyzing mRNA expression of tissue from aging or HGPS individuals across human, cynomolgus monkey, and mouse species, we confirmed that PAX5 was also highly expressed in various organs of naturally aged or HGPS individuals (Table EV1). We also confirmed the protein expression of PAX5 significantly increased in adipose stem cells isolated from old donors (Fig. 3D). These results strengthened the link between PAX5 overexpression and senescence/aging.

Next, we tested whether PAX5-mediated upregulation in LINE1 transcription is further enhanced in SIPS cells. ChIP assay revealed a higher enrichment of endogenous PAX5 at the LINE1 5′-UTR in SIPS cells than that in control cells (Fig. 3E, Appendix Fig. S2A,B). Subsequently, using RT-qPCR to measure the expression of endogenous LINE1, we found that overexpressing PAX5 could significantly promote the transcription of LINE1 in SIPS cells (Fig. 3F).

Considering that LINE1 derepression promotes cellular senescence, we hypothesized that PAX5 activating LINE1 and consequently accelerated cellular senescence. To test this hypothesis, we examined the expression of the Cyclin-Dependent Kinase Inhibitor p21, a widely used senescence marker, in X-ray induced SIPS, and found that overexpressing PAX5 significantly increased the expression of p21 in SIPS (Fig. 3G). In addition, β-gal staining, another widely used assay for measuring cellular senescence, revealed that overexpression of PAX5 increased the percentage of senescence-associated β-gal⁺ cells from ~50% to ~75% (Fig. 3H). Consequently, RT-qPCR revealed the mRNA level of two representative SASPs, IL1β and IL6, was significantly increased in PAX5 overexpressed SIPS cells (Fig. 3I). Moreover, to further validate that PAX5 promotes LINE1-mediated cellular senescence, we treated the irradiated cells with 3TC, a nucleoside reverse transcriptase inhibitor (NRTI) (Coates et al, 1992), and found that inhibiting LINE1 by 3TC negated the stimulatory effect of PAX5 on cellular senescence (Fig. 3J). These results suggested that PAX5 boosts LINE1 transcription and promotes cellular senescence.

## The longevity gene SIRT6 inhibits PAX5 transcription

PAX5 is typically highly expressed only in B-lymphoid lineage cells or some transformed cells, and is expressed at extremely low levels in other types of cells (Nutt et al, 1999; Torlakovic et al, 2006). Our surprising findings of upregulation in PAX5 expression during cellular senescence indicated that its transcription might be tightly suppressed under normal conditions, and the suppression might be disrupted in the context of cellular senescence. We therefore hypothesized that some senescence-associated factors that are involved in regulating LINE1 might play critical roles in suppressing PAX5 expression. SIRT6, a well-recognized longevity-associated gene, exhibits a decline in expression in senescent cells (Lee et al, 2020). Its occupancy on chromatin changes, and there may be an increased displacement from its binding sites to damage sites during aging (Kawahara et al, 2011; Tasselli et al, 2017). As a histone deacetylase, SIRT6 may regulate gene transcription as an epigenetic modifier. Thus, we proposed that SIRT6 might act as an upstream regulator of PAX5.

To investigate this, we first knocked down SIRT6 in cells and analyzed the expression level of PAX5. We observed increases in both the protein level and mRNA expression of PAX5 (Fig. 4A,B). Moreover, depleting SIRT6 led to the upregulation in PAX5 promoter activity and reintroduced SIRT6 in these cells inhibited the promoter activity (Fig. 4C). Furthermore, analysis of the transcriptome profile of the lung tissue of 86 donors ranging from 16 to 76 years old revealed a significant negative correlation between the mRNA expression levels of SIRT6 and PAX5 (Fig. 4D) (Lee et al, 2021; Data ref: Lee et al, 2021). This negative correlation also found in muscle, microglia, skin, testis and vagina (Fig. EV1) (Galatro et al, 2017; Xu et al, 2016; Data ref: Galatro et al, 2017;

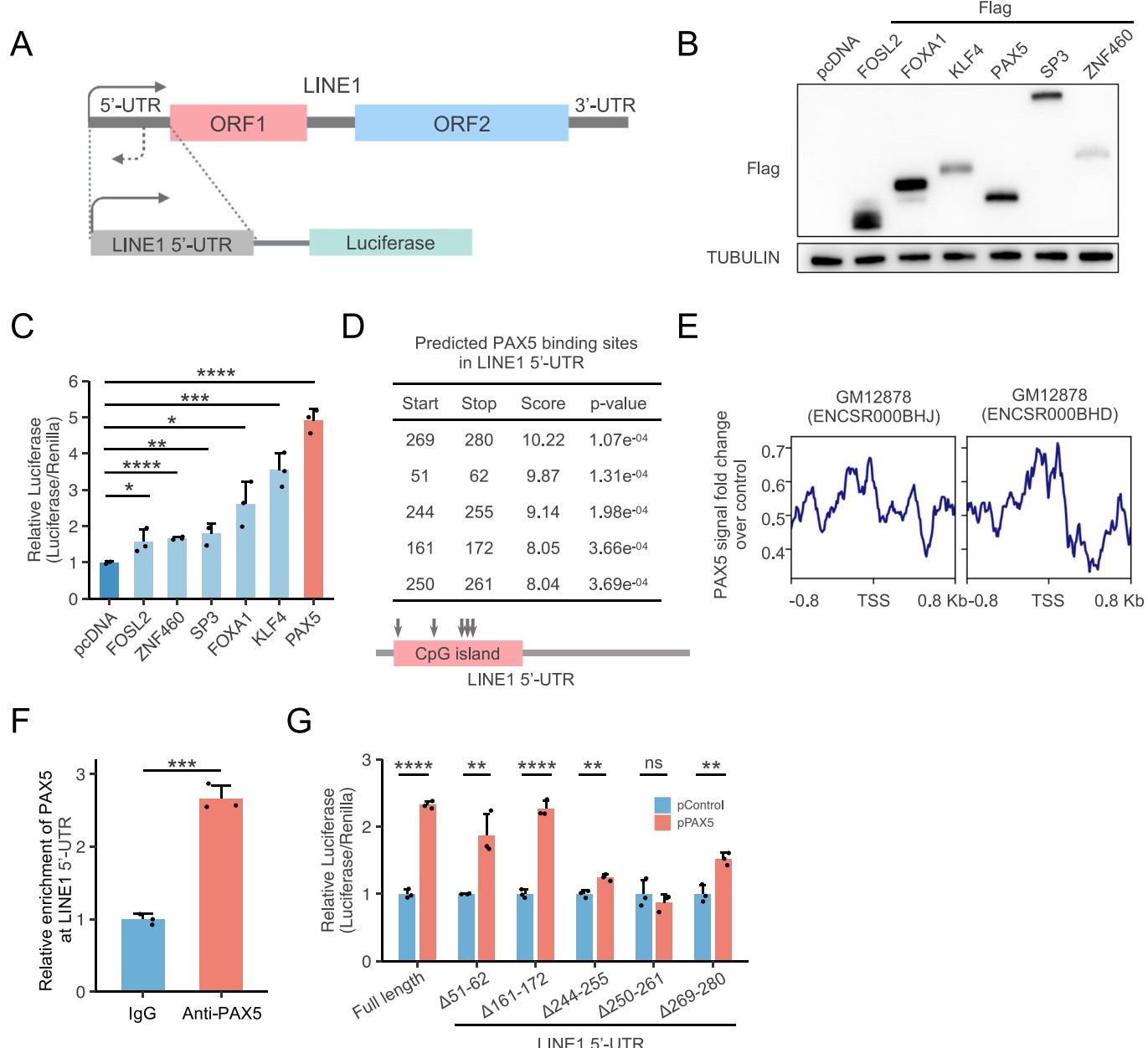

**Figure 2. PAX5 binds LINE1 5'-UTR and regulates LINE1 expression.**

(A) Schematic representation of the LINE1 5'-UTR and luciferase fusion reporter. (B) Expression of senescence associated upregulated LINE1 binding transcription factor candidates. (C) Transcriptional activity analysis of LINE1 5'-UTR when senescence associated upregulated LINE1 binding transcription factor candidates overexpressed; activity was measured by relative luciferase activity ($n = 3$, biological replicates). (D) FIMO results indicated potential PAX5 binding sites within the LINE1 5'-UTR. (E) PAX5 ChIP-Seq signal intensity within the LINE1 5'-UTR in GM12878 cell line. (F) ChIP analysis displayed endogenous PAX5 enrichment at the designated LINE1 5'-UTR sites ($n = 3$, biological replicates). (G) Transcriptional activity analysis of either full-length LINE1 5'-UTR or LINE1 5'-UTR with PAX5 binding site mutations (Δ51-Δ62/Δ161-Δ172/Δ244-255/Δ250-261/Δ269-Δ280) in the presence of PAX5 overexpression ($n = 3$, biological replicates). Error bars denote standard deviation (SD). Statistical significance was accessed by two-tailed Student's $t$ test. ns, not significant; $*P < 0.05$; $**P < 0.01$; $***P < 0.001$; $****P < 0.0001$. Source data are available online for this figure.

Xu et al, 2016). These results suggested that SIRT6 might regulate the expression of PAX5 through transcriptional mechanisms.

We then sought to determine whether the regulation of PAX5 by SIRT6 is direct at transcriptional level. We first analyzed SIRT6 ChIP-Seq data from H1-ESC cell lines provided by ENCODE, and the data mining revealed that SIRT6 had binding sites at the PAX5 promoter in

the normal H1-ESC cell line (Fig. 4E). These results were confirmed by ChIP assay, which indicates an enrichment of SIRT6 at PAX5 promoter region (Fig. 4F; Appendix Fig. S3A,B). In addition, when analyzing H3K27ac ChIP-Seq data from senescent IMR90 cells (Tasdemir et al, 2016; Data ref: Tasdemir et al, 2016), we observed a significant increase in the H3K27ac signal intensity—a substrate of

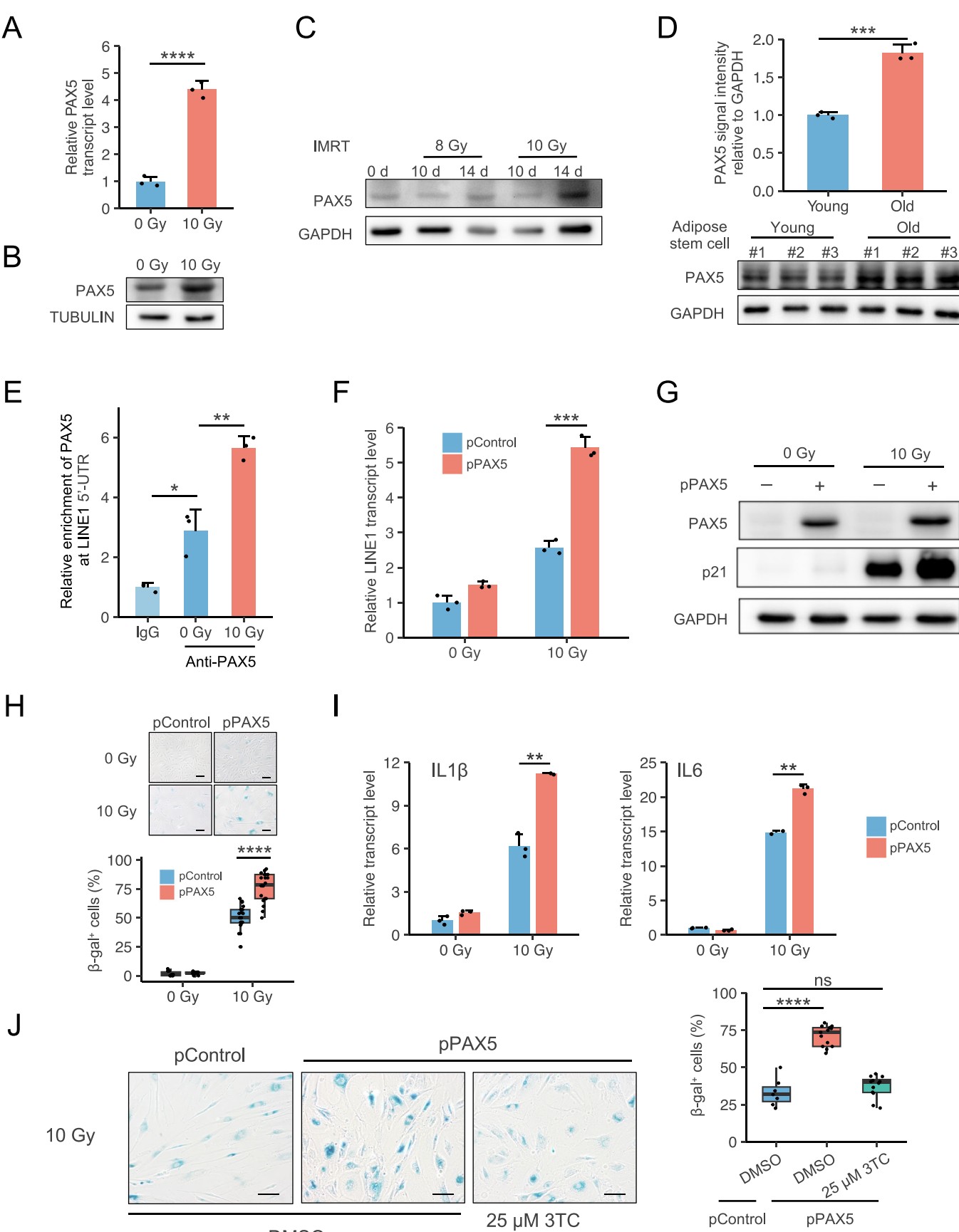

Figure 3. **PAX5 promotes cellular senescence via LINE1 activation.**

(A, B) HCA2-hTERT cells were exposed to 10 Gy X-ray irradiation and the relative PAX5 mRNA (A) and protein (B) expression were quantified at day 10 post-irradiation ($n = 3$, biological replicates). (C) IMR90-hTERT cells were exposed to 8 or 10 Gy X-ray irradiation, and the relative PAX5 protein expression was quantified by Western blot at day 10 or 14 post-irradiation. (D) Comparison of PAX5 protein expression levels between adipose stem cells isolated from young and old donors ($n = 3$, biological replicates). (E) ChIP analysis of endogenous PAX5 enrichment at the indicated LINE1 5'-UTR region in HCA2-hTERT cells collected on day 10 post 10 Gy X-ray irradiation ($n = 3$, biological replicates). (F) Relative LINE1 transcription level was quantified by RT-qPCR in PAX5-overexpressing stable line treated with 10 Gy X-ray irradiation and collected at day 10 post-irradiation ($n = 3$, biological replicates). (G–I) PAX5-overexpressing cells were treated with 10 Gy X-ray irradiation, and cells were harvested on day 10 post-irradiation. (G) Cellular senescence was assessed by quantifying the expression of the senescence marker p21 using Western blot. (H) β-gal$^+$ cells were quantified using SA-β-gal staining ($n = $ at least 10, biological replicates). (I) Transcript expression of representative SASP genes, IL1β and IL6, was quantified by RT-qPCR ($n = 3$, biological replicates). (J) PAX5-overexpressing cells exposed to 10 Gy X-ray irradiation were treated with DMSO or 25 μM 3TC for 10 days. Cellular senescence was assessed by quantifying β-gal$^+$ cells using SA-β-gal staining ($n = $ at least 7, biological replicates). For all boxplots, the minimum and maximum are typically 1.5 times the Interquartile Range (IQR) from the quartiles, the center value is the median, the box edges are the 25th (Q1) and 75th (Q3) percentiles, and the whiskers extend to the nearest non-outlier points within 1.5 times the IQR. Error bars denote SD. Statistical significance was accessed by two-tailed Student's $t$ test. ns, not significant; *$P < 0.05$; **$P < 0.01$; ***$P < 0.001$; ****$P < 0.0001$ (Scale bar: 100 μm). Source data are available online for this figure.

SIRT6—within the PAX5 promoter region (Fig. 4G), suggesting that SIRT6 may inhibit PAX5 expression by deacetylating histone modifications in the PAX5 promoter region, and this regulation might be abrogated in senescent cells. Therefore, we hypothesized that the lack of SIRT6-mediated suppression on PAX5 transcription might lead to the senescence-associated upregulation in PAX5 mRNA and protein level. ChIP assay revealed a lower enrichment of SIRT6 at the PAX5 promoter in SIPS cells than that in control cells (Fig. 4H; Appendix Fig. S3C). And SIRT6 overexpression completely abrogated the senescence-associated upregulation in PAX5 mRNA level in senescent cells (Fig. 4I). These findings indicated that PAX5 is negatively regulated by SIRT6.

To demonstrate whether SIRT6 regulates LINE1 transcription and cellular senescence through suppressing PAX5 transcription, we overexpressed SIRT6 and/or PAX5 in cells and used a LINE1 5'-UTR and luciferase fusion reporter to evaluate the transcriptional activity of the LINE1 5'-UTR. The results showed that over-expression of exogenous PAX5 counteracted SIRT6-mediated inhibition of LINE1 transcription (Fig. 4J), and was further confirmed by β-gal staining (Fig. 4K). Since previous studies demonstrated that the expression of SIRT6 declines in senescent cells (Mao et al, 2012; Lee et al, 2020), our data indicate that the PAX5-mediated LINE1 activation and cellular senescence may be a consequence of the decreased activity of SIRT6.

LINE1, which comprises a substantial proportion of the human genome, is progressively derepressed and aberrantly transcribed during cellular senescence and aging. This activation of LINE1 provokes a type I interferon response and instigates the secretion of SASP factors. The paracrine effects of SASP subsequently cascade and amplify cellular senescence, thereby escalating the level of chronic inflammation within the body and shortening the lifespan. Consequently, gaining a comprehensive understanding of LINE1's regulatory mechanisms is crucial for advancing research into aging and cellular senescence. Our study presents a method of leveraging transcriptome sequencing data from various cellular senescence to identify senescence-related, upregulated LINE1 at the locus level. We then forecasted novel transcription factors capable of activating these LINE1 in senescent cells. We screened potential transcription factors by a LINE1 5'-UTR and luciferase fusion reporter and successfully identified a new transcription factor, PAX5, that might regulate LINE1 transcription.

The mechanisms of senescence are remarkably complicated, and universal markers for senescent identification remain elusive

(Hernandez-Segura et al, 2018; Sonawane et al, 2017). Furthermore, the expression and occupancy of transcription factors may be tissue or cell-type specific. Even FOXA1, which has been reported to be upregulated in senescent cells, was found to be significantly decreased in SIPS and RS IMR90 cells (Li et al, 2013; Casella et al, 2019). Given this, we hypothesize that the transcription factors regulating LINE1 transcription may also exhibit this specificity. Our approach, which forecasts senescence-associated upregulated LINE1 regulating transcription factors and then screens them using a LINE1 5'-UTR and luciferase fusion reporter, could potentially be used to identify more tissue-specific transcription factors that regulate LINE1 and promote cellular senescence.

Apart from LINE1, other transposable elements could also promote cellular senescence. It has been reported that the upregulation of endogenous retroviruses such as HERVK can activate innate immunity and trigger cellular senescence (Liu et al, 2023). Our method could adapt the luciferase fusion reporter according to the type of transposable elements under study, allowing for swift screening of new regulating transcription factors. Furthermore, when studying diseases caused by the abnormal activation of transposable elements, researchers could leverage transcriptome sequencing data derived from public databases to predict the transcription factors regulating these elements (Evering et al, 2023; Valdebenito-Maturana et al, 2022). Our method could expedite the identification of transcription factors that regulate disease-related transposable elements, based on public transcriptome data.

At present, the most widely studied drugs targeting LINE1 derepression and activating immune response process are NRTIs. Multiple studies have demonstrated that NRTIs can mitigate the type I interferon response triggered LINE1 cDNA accumulation and enhance the health and extend lifespan of premature mice with LINE1 derepression (Cecco et al, 2019; Simon et al, 2019). Although NRTIs have demonstrated promising therapeutic outcomes, their side effects should not be overlooked. Some research has reported that NRTIs could affect telomerase activity, and their off-target effects increase the risk of normal cells undergoing cellular senescence (Leeansyah et al, 2013; Hukezalie et al, 2012). Furthermore, NRTIs could also inhibit the mitochondrial-specific DNA polymerase-γ, affecting the synthesis and accuracy of mitochondrial DNA (Smith et al, 2013). Moreover, the accumulation of LINE1 RNA can lead to SUV39H1 activity supression, heterochromatin erosion and the onset of senescence phenotypes (Valle et al, 2022). Whether NRTIs affect LINE1 transcription warrants

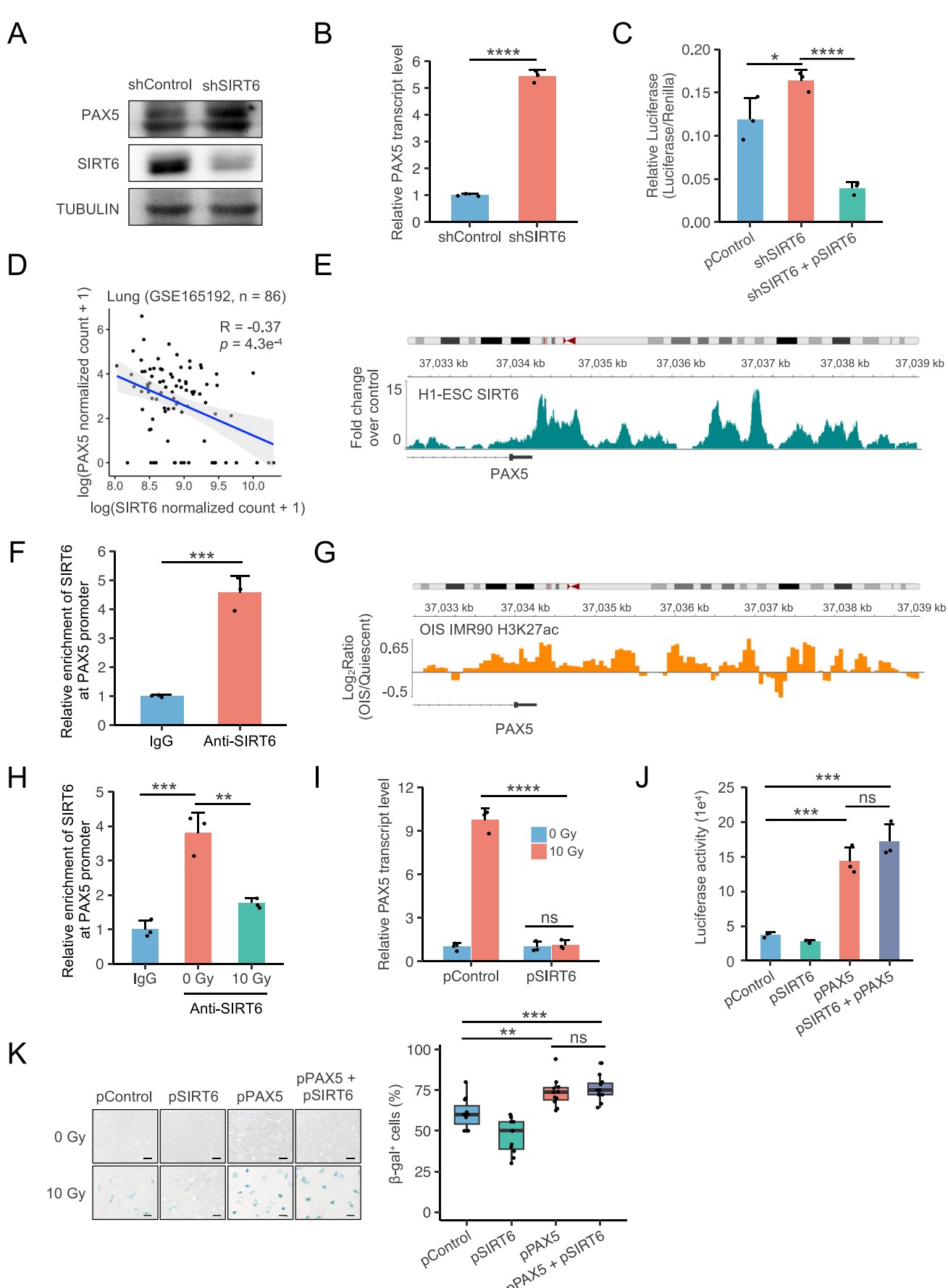

**Figure 4.   SIRT6 directly regulates PAX5 transcription.**

(A, B) Quantification of PAX5 protein expression in SIRT6-depleted cells (A) by Western blot and transcript expression (B) by RT-qPCR ($n = 3$, biological replicates).
(C) Evaluation of the transcriptional activity of the PAX5 promoter in SIRT6-depleted and SIRT6-reintroduced cells, quantified by luciferase activity using a PAX5 promoter
and luciferase fusion reporter ($n = 3$, biological replicates). (D) Pearson correlation of DESeq2 normalized counts of SIRT6 and PAX5 of lung tissue of 86 donors ranging
from 16 to 76 years old. (E) Genomic tracks display ChIP-Seq data for SIRT6 surrounding the PAX5 promoter region in H1-ESC cells. (F) ChIP analysis showed endogenous
SIRT6 enrichment in PAX5 promoter in HCA2-hTERT cells ($n = 3$, biological replicates). (G) Genomic tracks display ChIP-Seq data for H3K27ac surrounding the PAX5 promoter
promoter region in OIS IMR90 cells. (H) ChIP analysis of SIRT6 enrichment at PAX5 promoter in HCA2-hTERT cells collected on day 10 post 10 Gy X-ray irradiation ($n = 3$,
biological replicates). (I) Relative PAX5 transcript expression, quantified by RT-qPCR, in SIRT6-overexpressing cells treated with 10 Gy X-ray irradiation and collected on
day 14 post-irradiation ($n = 3$, biological replicates). (J) Analysis of the transcriptional activity of the LINE1 5'-UTR in response to SIRT6 and/or PAX5 overexpression,
quantified by luciferase activity using a LINE1 5'-UTR and luciferase fusion reporter ($n = 3$, biological replicates). (K) PAX5 or/and SIRT6 overexpressed cells were exposed
to 10 Gy X-ray irradiation and collected at day 10 post-irradiation. Cellular senescence was accessed by quantifying β-gal$^+$ cells using SA-β-gal staining ($n = 12$, biological
replicates). For the boxplot, the minimum and maximum are typically 1.5 times the Interquartile Range (IQR) from the quartiles, the center value is the median, the box
edges are the 25th (Q1) and 75th (Q3) percentiles, and the whiskers extend to the nearest non-outlier points within 1.5 times the IQR. Error bars denote SD. Statistical
significance was accessed by two-tailed Student's $t$ test. ns, not significant; *$P < 0.05$; **$P < 0.01$; ***$P < 0.001$; ****$P < 0.0001$ (Scale bar: 100 μm). Source data are available
online for this figure.

further investigation. The transcription factor PAX5, which we identified through our screening process, could potentially activate LINE1 transcription by targeting it. Therefore, it could present a more effective drug target for inhibiting LINE1 and treating senescence-associated diseases. While SIRT6, SIRT7, and BMAL1 can all inhibit LINE1 activation and may serve as potential targets for suppressing LINE1 transcription, the challenge of developing activators is considerably greater than that of developing inhibitors. Therefore, the engineering of inhibitors targeting PAX5 may present a less difficult endeavor. Notably, since PAX5 is only highly expressed in B-lymphoid lineage and senescent cells, this high degree of tissue specificity may minimize potential side effects.

In summary, our work established a method based on the transcript of LINE1 commonly upregulated in cellular senescence to identify enriched transcription factors and screen candidate transcription factors using a LINE1 5'-UTR and luciferase fusion reporter. We discovered a novel transcription factor, PAX5, that regulates LINE1 transcriptional activity. Our mechanistic studies indicated that PAX5 promotes cellular senescence by binding to the LINE1 5'-UTR and enhancing LINE1 transcription. We also demonstrated that SIRT6 could bind to the PAX5 promoter and inhibit its transcription.

# Methods

## Cell culture

All cell lines used in the study were cultured in DMEM medium supplemented with 10% fetal bovine serum (Gibco, Cat. #16000-044), 1% nonessential amino acid (Gibco, Cat. #11140-050), and 1% penicillin/streptomycin (Gibco, Cat. #15140-122). The cultures were maintained in a 5% $CO_2$ incubator at 37 °C. The adipose stem cell was isolated and maintained (Zhang et al, 2020) and used in this study following the approval of the Ethics Committees of Tongji University and Changzheng Hospital (Approval No. 2015yxy111).

## Plasmids, antibodies, and primers

The ORF of PAX5 was synthesized by General Biol (Anhui, China) and inserted into pEGFP-N1 backbone. All mutants were generated by the site-specific mutagenesis method (Transgen, FM111-02) and were confirmed by sequencing.

The antibodies used in the study are as follows: PAX5 (Abcam, Cat. #ab109443), SIRT6 (CST, Cat. #12486), Flag (Abclonal, Cat. #AE005), GAPDH (Proteintech, Cat. #HRP-60004), β-TUBULIN (Bioworld, Cat. #AP0064), p21 (Abcam, Cat. #ab109199).

The primers used in this study are listed in Appendix Table S1.

## Generation of PAX5 overexpression stable cell line in HCA2-hTERT

pCMV-PAX5-Flag conferring resistance to G418 was transfected into HCA2-hTERT cells using the Lonza 4D electroporation machine with DT-130 program. At 24 h post transfection, 1 mg/ml G418 was applied to the transfected cells for selection. On days 10–14, colonies were picked and expanded for Western blot analysis.

## Transfections

All HCA2-hTERT derived cell lines were electroporated on a Lonza 4D machine with DT-130 program. HEK293 cells were transfected with the indicated plasmids using polyethylenimine transfection.

## β-gal staining

Cells were irradiated with X-ray generated by an RS2000Pro source to induce senescence. Ten days later, the cells were fixed in 2% formaldehyde and 0.2% glutaraldehyde in PBS for 5 min at room temperature and washed twice with PBS. Subsequently, the staining solution (1 mg/mL X-gal in dimethylformamide, 40 mM citric acid/sodium phosphate buffer, 5 mM potassium ferrocyanide, 5 mM potassium ferricyanide, 150 mM sodium chloride, and 2 mM magnesium chloride) was added, and the cells were incubated overnight at 37 °C. Images of the stained cells were taken with a bright-field microscope.

## ChIP assay

The ChIP assay was performed in HCA2-hTERT cells as previously reported (Chen et al, 2019). In brief, to quantify the enrichment of endogenous PAX5 on LINE1 5'-UTR, HCA2-hTERT cells were

collected for ChIP assay using anti-PAX5 antibody. To quantify the enrichment of endogenous SIRT6 on PAX5 promoter, HCA2-hTERT cells were collected for ChIP assay using anti-SIRT6 antibody. DNA purified after immuno-precipitate and input samples were used as templates for RT-qPCR quantification. RT-qPCR was performed in a ViiA 7 real-time PCR system (Applied Biosystems). The primers used in this study is listed in Appendix Table S1.

## Luciferase assay

The PAX5 promoter was cloned from genomic DNA extracted from HEK293 cells, and a 5000 bp DNA segment upstream of the transcription start site was cloned into the pGL3-Basic plasmid. The mutated LINE1 5'-UTR promoter that lacked the PAX5 binding site, was generated by site-specific mutagenesis based on the above plasmids and confirmed by sequencing. Luciferase assay for quantification of LINE1 transcriptional activity was performed in HeLa cells, which was seed at a density of $1 \times 10^5$ cell per well in a six-well plate, 24 h later cells were cotransfected using the CN114 program in a Lonza 4D instrument (Lonza) with indicated plasmids and Renilla luciferase, an internal reference to normalize the transfection efficiency. Twenty-four hours after transfection, cells were collected and lysed. Luciferase assay for quantification of PAX5 activity was performed in HCA2-hTERT cells, and transfected 48 h after seeding with DT130 program in a Lonza 4D instrument, another 24 h later, cells were collected and lysed. Subsequently, luciferase activity was measured with a dual-luciferase reporter system (Promega, E1910) and a GloMax luminometer (Promega, E5311).

## Prediction of transcription factors binding to upregulated LINE1 regions in various types of senescent cells

RNA-Seq data for OIS, RS, and SIPS WI-38 cells were retrieved from the Gene Expression Omnibus (GSE130727). Quality control was conducted using FastQC v0.11.9. Adapters and low-quality reads were filtered out using Cutadapt v2.8. The samples were then aligned to the GRCh38 genome with STAR v2.5.3a using parameters "--outFilterMultimapNmax 100 --winAnchorMultimapNmax 100 -alignEndsType EndToEnd -alignEndsProtrude 100 DiscordantPair --outFilterScoreMinOverLread 0.4 --outFilterMatchNminOverLread 0.4". Quantification of repetitive elements was achieved through SQuIRE v0.9.9.92. Differentially expressed repetitive elements at the locus level were determined using DESeq2 v1.16.1. LINE1 variants that were differentially upregulated (with fold change >1.5 and adjusted $p$-value < 0.01) at least in two senescence types were identified as commonly upregulated LINE1 variants. Sequences ranging from −200 bp to 1000 bp of these commonly upregulated LINE1 variants were extracted from GRCh38 genome using the 'getfasta' function in bedtools v2.31.0. These sequences served as input for XSTREME v5.5.4 and using parameters "Shuffled input sequences Minimum width 8 Maximum width 15" to predict enriched transcription factors. The SEA v5.5.4 with parameters "--thresh 0.05 --order 2 --seed 0 --align center --motif-pseudo 0.01 --m JASPAR2022_CORE_vertebrates_non-redundant_v2.meme" was used to perform motif enrichment analysis in these sequences. FIMO with parameters "Match $p$-value < 1e$^{-4}$ motif id MA0014.3" were used to predict PAX5 binding sites on the LINE1 5'-UTR based on its binding motif.

## ChIP-Seq analysis

ChIP-Seq data for PAX5 and SIRT6 were sourced from ENCODE (ENCSR000BHJ and ENCSR000BHD correspond to PAX5; ENCSR000AUS corresponds to SIRT6). Coordinates for all full-length, intact LINE1 variants were retrieved from L1Base (Penzkofer et al, 2017). The 'bamCompare' function in deepTools v3.5.4 was utilized to compute the fold change in PAX5 signal relative to the control. ChIP-Seq data for H3K27ac in OIS IMR90 cells were acquired from the Gene Expression Omnibus (GSE74238). The 'bigwigCompare' function in deepTools was employed to determine the Log2Ratio of the H3K27ac signal value at the PAX5 promoter compared to quiescent cells. Visualization of genomic tracks showcasing ChIP-Seq data was facilitated by the IGV-Web app v1.13.11.

# Data availability

This study includes no data deposited in external repositories.

The source data of this paper are collected in the following database record: biostudies:S-SCDT-10_1038-S44319-024-00176-9.

# Peer review information

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

## Acknowledgements

The work was supported by the National Key R&D Program of China (2022YFA1103703 and 2021YFA1102003 to ZM), China Postdoctoral Science Foundation (2023M732662 to HT), the National Natural Science Foundation of China (Grant Nos. 82225017, 32270750 and 82071565 to ZM, 32171288 to YJ, 32200603 to WZ, 32100605 to AG), the Shanghai Sailing Program (22YF1434500 to WZ and 21YF1436000 to AG).

## Author contributions

**Huanyin Tang**: Conceptualization; Data curation; Formal analysis; Funding acquisition; Investigation; Visualization; Methodology; Writing—original draft; Project administration. **Jiaqing Yang**: Conceptualization; Data curation; Formal analysis; Validation; Visualization; Methodology; Writing—original draft. **Junhao Xu**: Data curation; Formal analysis; Validation. **Weina Zhang**: Formal analysis; Funding acquisition; Validation. **Anke Geng**: Formal analysis; Funding acquisition; Validation. **Ying Jiang**: Conceptualization; Funding acquisition; Investigation. **Zhiyong Mao**: Conceptualization; Resources; Supervision; Funding acquisition; Investigation; Methodology; Writing—original draft; Project administration; Writing—review and editing.

Source data underlying figure panels in this paper may have individual authorship assigned. Where available, figure panel/source data authorship is listed in the following database record: biostudies:S-SCDT-10_1038-S44319-024-00176-9.

## Disclosure and competing interests statement

The authors declare no competing interests.

# Expanded View Figure

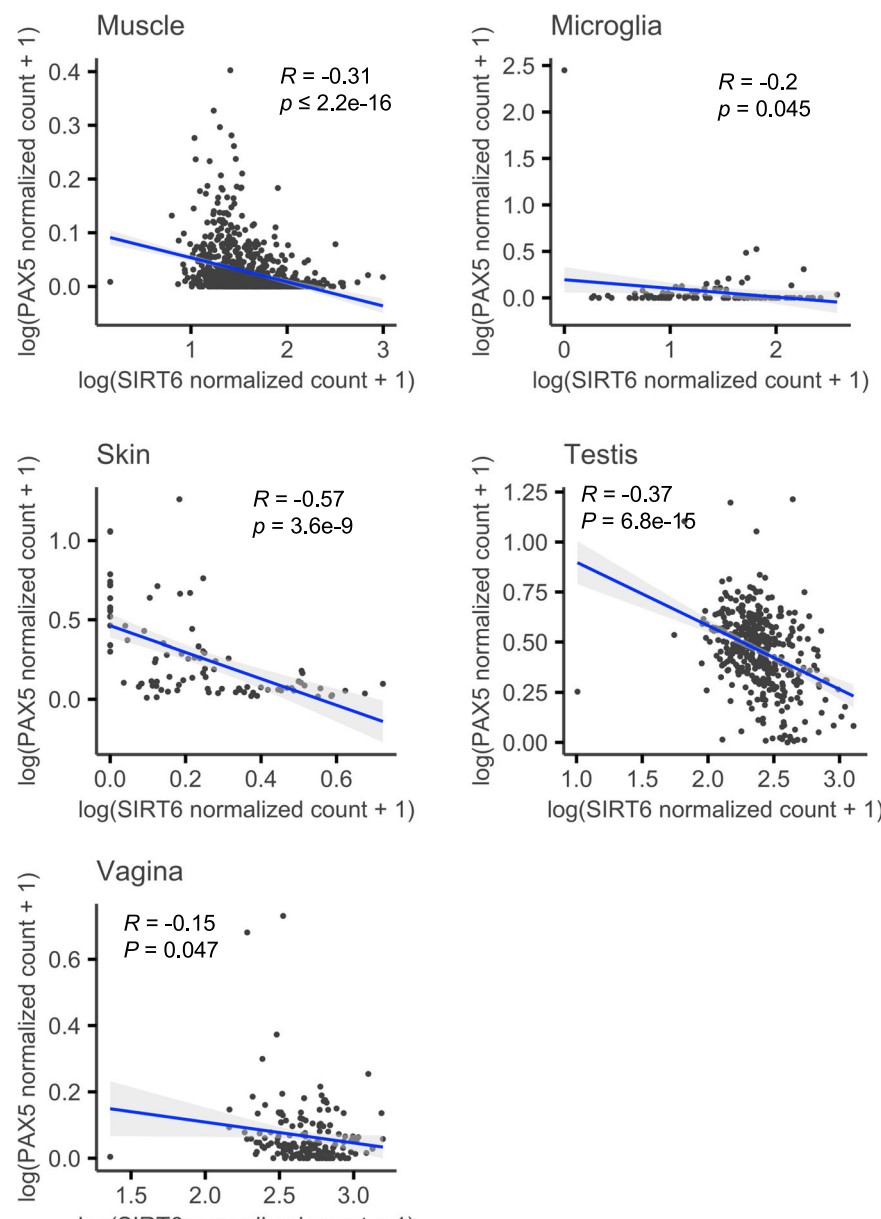

**Figure EV1. SIRT6 expression is negatively correlated with PAX5 in multiple tissues.**

Pearson correlation of DESeq2 normalized counts of SIRT6 and PAX5 of muscle (GTEx, $n = 881$), microglia (GSE99074, $n = 71$), skin (GSE85861, $n = 91$), testis (GTEx, $n = 410$) and vagina (GTEx, $n = 173$) tissue.

