## [Peer Review File · EMBO Reports]

The transcription factor PAX5 activates human LINE1 retrotransposons to induce cellular senescence

Huanyin Tang, Jiaqing Yang, Junhao Xu, Weina Zhang, Anke Geng, Ying Jiang, and Zhiyong Mao

Corresponding author(s): Zhiyong Mao (zhiyong_mao@tongji.edu.cn)

Review Timeline:

Submission Date:	5th Jan 24
Editorial Decision:	26th Jan 24
Revision Received:	24th Apr 24
Editorial Decision:	16th May 24
Revision Received:	22nd May 24
Accepted:	3rd Jun 24

Editor: Esther Schnapp

Transaction Report:

Dear Dr. Mao,

Thank you for the submission of your manuscript to EMBO reports. We have now received the full set of referee reports that is pasted below.

As you will see, the referees acknowledge that the findings are potentially interesting. However, they also raise a few concerns, and I think that all should be addressed, especially the points raised by referee 3 are important. Please let me know if you have any comments or questions regarding the reports.

I would thus like to invite you to revise your manuscript with the understanding that the referee concerns must be fully addressed and their suggestions taken on board. Please address all referee concerns in a complete point-by-point response. Acceptance of the manuscript will depend on a positive outcome of a second round of review. It is EMBO reports policy to allow a single round of major revision only and acceptance or rejection of the manuscript will therefore depend on the completeness of your responses included in the next, final version of the manuscript.

We realize that it is difficult to revise to a specific deadline. In the interest of protecting the conceptual advance provided by the work, we recommend a revision within 3 months (27th Apr 2024). Please discuss the revision progress ahead of this time with the editor if you require more time to complete the revisions.

- 1) A data availability section providing access to data deposited in public databases is missing. If you have not deposited any data, please add a sentence to the data availability section that explains that.
- 2) Your manuscript contains statistics and error bars based on $n=2$. Please use scatter blots in these cases. No statistics should be calculated if $n=2$.

5) a complete author checklist, which you can download from our author guidelines <<https://www.embopress.org/page/journal/14693178/authorguide>>. Please insert information in the checklist that is also reflected in the manuscript. The completed author checklist will also be part of the RPF.

6) Please note that all corresponding authors are required to supply an ORCID ID for their name upon submission of a revised manuscript (<<https://orcid.org/>>). Please find instructions on how to link your ORCID ID to your account in our manuscript tracking system in our Author guidelines <<https://www.embopress.org/page/journal/14693178/authorguide#authorshipguidelines>>

7) Before submitting your revision, primary datasets produced in this study need to be deposited in an appropriate public database (see <https://www.embopress.org/page/journal/14693178/authorguide#datadeposition>). Please remember to provide a reviewer password if the datasets are not yet public. The accession numbers and database should be listed in a formal "Data Availability" section placed after Materials & Method (see also <https://www.embopress.org/page/journal/14693178/authorguide#datadeposition>). Please note that the Data Availability Section is restricted to new primary data that are part of this study. * Note - All links should resolve to a page where the data can be accessed. *
If your study has not produced novel datasets, please mention this fact in the Data Availability Section.

I look forward to seeing a revised form of your manuscript when it is ready.

Yours sincerely,

Referee #1:

Tang and Yang et al. identify PAX5 as a regulator of L1 expression active in senescent cells. A pool of candidate regulators is identified from motif analysis of L1 copies upregulated in senescent cells. PAX5 is then identified as the most promising factor and shown to induce L1 expression, SASP and increase senescence. The effect on senescence could be rescued by 3TC treatment, showing it is mediated by (L1) reverse transcription. The authors additionally show that PAX5 upregulation during senescence seems to depend on reduced SIRT6 activity. The manuscript is well written, concise and interesting. Most of my comments are minor clarifications.

The sentence on line 74 describing the autocrine, juxtacrine and paracrine effects of SASP could be made clearer.

It is unclear what the columns in fig 1a represent. Are these biological replicates, 2 per condition? If so, one sample in the SIPS condition and one in the OIS condition appear to express L1s at much higher level than the other biological replicate, which is interesting. This suggests some heterogeneity in L1 activation following senescence. Please include a brief description of the RNA-seq samples (ideally around line 138) and label the axes of figure 1a.

Which L1 families do the upregulated variants belong to? Are these all full-length L1s or are some of these truncated?

Line 160 referring to figure 1e does not seem accurate: PAX5 and SP3 motif were more likely to have single matches than ZNF460 and FOSL2, but for all of these, one match was the most common occurrence. Additionally, PAX5 and SP3 motifs were found in more than one copy in multiple L1s. Please correct the phrasing. Comparisons would also be easier if the subpanels of figure 1e had matching x axes.

On line 265 "the loss function of sirt6" should be corrected to "decreased function" or "decreased activity"

Please include the configurations/arguments used for STAR, STREME, SEA AND FIMO in the methods section.

Please describe the primers utilized to conduct the L1 RT-qPCR for both target and reference.

Vera Gorbunova

Referee #2:

Using different approaches, Tang et al. identified a novel transcription factor PAX5 that binds to 5'-UTR and promotes LINE-1 transcription, and they reported that this regulation is suppressed in young cells while during cellular senescence more PAX5 is enriched to LINE-1 5'-UTR and activates LINE-1 transcription. Additionally, the authors demonstrated that SIRT6 inhibits PAX5 transcription to suppress the onset of cellular senescence. Overall, this is an interesting and well-designed work. This manuscript would be a nice addition to EMBO reports and the field of aging. However, before I can recommend the publication, some concerns need to be addressed.

1. In 2019, both Sedivy lab and Gorbunova lab demonstrated that the transcriptional activation of LINE1 could be the driver of cellular senescence and aging. Besides PAX5, the authors confirmed their finding of FOXA1 and SIRT6 regulating LINE1. Can authors also examine whether SIRT6 represses FOXA1 mediated activation of LINE1 transcription using the luciferase assay? If so, SIRT6 might be the common upstream of both of the transcription factors - PAX5 and FOXA1.
2. Through data mining, the authors showed that there are 96 upregulated LINE1 variants in different types of senescent cells. Can the authors quantify the percentage of these variants in all LINE1 elements?
3. More details should be added to the legends. For instances, the abbreviation of RS, SIPS and OIS should be explained in the legends, and this would help readers understand the paper better.
4. Throughout the manuscript, some sentences are hard to understand. Here are some that need to be corrected.
 - As in LINE 71, SASP has been explained, there is no need to show the full name of the phrase again in Line 213-214.
 - In LINE 250, the reference is missing.
 - In LINE 253, the sentence needs to be rephrased.
 - In LINE 265, the sentence needs to be rephrased.
 - In Figure 1C, the table should be re-organized.
 - The spaces are missing before the parentheses that contain the references.

Referee #3:

I have looked at the manuscript and I indeed have only minor comments, as the other reviewers do, which should/could be corrected and addressed in minor revisions.

My three 'main' comments (and one very minor, number 4) would be:

1. Most of the work that is performed in the ageing experiments is done with overexpressed proteins. For example, in Figure 2F, the Pax5 ChIP enrichment on LINE 5'UTRs are done with GFP-overexpressed Pax5 fusion. Similarly, the SIRT6 ChIP data in Figure 4F is performed with a GFP-fusion protein overexpressed in HEK293 cells. The 'only' endogenous data the authors present is provided from published data: e.g. encode (H1-ESC cell line for SIRT6) and other not necessarily relevant cell lines (for PAX5). The authors should really attempt to validate, at the very least, the binding of endogenous proteins (PAX5 and/or SIRT6) in their aged cells. Otherwise, the arguments of 'direct' regulation are done in completely artificial setups with regards to the ageing phenotype. Since chromatin regulation is likely to be cell-type specific, I think this is a major point the authors should address.

2. Related to their ChIP data, I think key controls are missing. They are easy to address though, and the authors may already have the material: the negative controls for the ChIP are currently done using chromatin of cells that express the empty GFP plasmid. To truly conclude enrichment, the authors should use, in addition, a negative control from the same chromatin (e.g. qPCR analysis of a negative, not-bound target gene of Pax5 and Sirt6 in cells overexpressing the fusion protein Pax5-GFP or Sirt6-GFP), otherwise it is impossible to ascertain if what they see is genomic background.

3. The conclusion on Pax5 mediates cellular senescence via LINE1 transcription should be toned down and/or changed. The authors show that inhibiting RT activity in Pax5 overexpressing conditions attenuates molecular (Beta gal) features of senescence, but they do not show that inhibiting LINE1 transcription does so (Figure 3).

4. The authors should specify from early on and on abstract that they are studying human LINE elements.

Referee #1:

Tang and Yang et al. identify PAX5 as a regulator of L1 expression active in senescent cells. A pool of candidate regulators is identified from motif analysis of L1 copies upregulated in senescent cells. PAX5 is then identified as the most promising factor and shown to induce L1 expression, SASP and increase senescence. The effect on senescence could be rescued by 3TC treatment, showing it is mediated by (L1) reverse transcription. The authors additionally show that PAX5 upregulation during senescence seems to depend on reduced SIRT6 activity. The manuscript is well written, concise and interesting. Most of my comments are minor clarifications.

We are delighted by the reviewer's positive comments.

1. The sentence on line 74 describing the autocrine, juxtacrine and paracrine effects of SASP could be made clearer.

We appreciate the suggestion and have rephrased the description of the autocrine, juxtacrine, and paracrine effects of SASP in the revised manuscript. The revised section is presented as follows: Previous studies have laid the groundwork for understanding how LINE1 retrotransposons can trigger SASP. SASP may induce senescence in an autocrine fashion, reinforcing senescence within the originating cell (Tasdemir & Lowe, 2013; Campisi, 2013). Additionally, these factors are involved in inducing senescence in neighboring cells, which is called "paracrine senescence". Furthermore, SASP factors can also participate in juxtacrine signaling transduction by directly interacting with adjacent cells to propagate senescence, thereby amplifying the cellular senescence signaling cascade (Acosta et al, 2013; Hoare et al, 2016).

2. It is unclear what the columns in fig 1a represent. Are these biological replicates, 2 per condition? If so, one sample in the SIPS condition and one in the OIS condition appear to express L1s at much higher level than the other biological replicate, which is interesting. This suggests some heterogeneity in L1 activation following senescence. Please include a brief description of the RNA-seq samples (ideally around line 138) and label the axes of figure 1a.

We thank the reviewer for the suggestion. We have added a description of the RNA-seq samples and have labeled the axes in Figure 1A. The added description is presented as follows: There are two replicates in each type of cellular senescence.

3. Which L1 families do the upregulated variants belong to? Are these all full-length L1s or are some of these truncated?

Pie Chart of Element Counts

We thank the reviewer for the suggestion. We have analyzed the common up-regulated LINE1 variants across different L1 families. In order to identify the LINE1 variants, we used L1Base2 for annotation and found that 13.84% of LINE1 variants are over 6000 bp, while 9.3% are between 4500 and 6000 bp with regulatory potential. The remainder are truncated LINE1s, indicating widespread activation of different LINE1 variants in senescent cells.

4. Line 160 referring to figure 1e does not seem accurate: PAX5 and SP3 motif were more likely to have single matches than ZNF460 and FOXL2, but for all of these, one match was the most common occurrence. Additionally, PAX5 and SP3 motifs were found in more than one copy in multiple L1s. Please correct the phrasing. Comparisons would also be easier if the subpanels of figure 1e had matching x axes.

We thank the reviewer for the suggestion. We have rephrased the text and updated Figure 1E with subpanels that have matching X axes to enhance comparability. The revised section is presented as follows: The occurrence of a single match was the most common. Compared to motifs of FOSL2 and ZNF460, those of SP3, KLF4, YY1, PAX5, and FOXA1 were likely to appear once. Furthermore, the motifs of PAX5 and SP3 exhibited multiple matches across multiple LINE1 variants (Fig. 1E).

5. On line 265 "the loss function of sirt6" should be corrected to "decreased function" or "decreased activity"

We thank the reviewer for pointing out the issues. Following the suggestion, we have corrected the term "the loss function of SIRT6" to "decreased activity of SIRT6".

6. Please include the configurations/arguments used for STAR, STREME, SEA AND FIMO in the methods section.

We appreciate the reviewer's comments and have included the parameters for STAR, STREME, SEA, and FIMO in the methods section.

7. Please describe the primers utilized to conduct the L1 RT-qPCR for both target and reference.

We thank the reviewer for the suggestion. The primers we used in RT-qPCR are listed in the new Appendix Table S1.

Vera Gorbunova

Referee #2:

Using different approaches, Tang et al. identified a novel transcription factor PAX5 that binds to 5'-UTR and promotes LINE-1 transcription, and they reported that this regulation is suppressed in young cells while during cellular senescence more PAX5 is enriched to LINE-1 5'-UTR and activates LINE-1 transcription. Additionally, the authors demonstrated that SIRT6 inhibits PAX5 transcription to suppress the onset of cellular senescence. Overall, this is an interesting and well-designed work. This manuscript would be a nice addition to EMBO reports and the field of

aging. However, before I can recommend the publication, some concerns need to be addressed.

We thank the reviewer for the positive comments.

1. In 2019, both Sedivy lab and Gorbunova lab demonstrated that the transcriptional activation of LINE1 could be the driver of cellular senescence and aging. Besides PAX5, the authors confirmed their finding of FOXA1 and SIRT6 regulating LINE1. Can authors also examine whether SIRT6 represses FOXA1 mediated activation of LINE1 transcription using the luciferase assay? If so, SIRT6 might be the common upstream of both of the transcription factors - PAX5 and FOXA1.

We appreciate the suggestion. To demonstrate whether SIRT6 regulates LINE1 transcription through suppressing FOXA1 transcription, we overexpressed SIRT6 and/or FOXA1 in cells and used the LINE1 5'-UTR and luciferase fusion reporter to evaluate the transcriptional activity of the LINE1 5'-UTR. The results showed that FOXA1 overexpression cannot rescue the suppressive effect of SIRT6 on LINE1 transcription. This indicates that SIRT6 likely operates through a different mechanism and may not act as a common upstream regulator for both PAX5 and FOXA1 in this context. Since it is out of the scope of this study, we have chosen not to include this data in this manuscript. But if the reviewer feels it necessary, we may include it in the further revision.

2. Through data mining, the authors showed that there are 96 upregulated LINE1 variants in different types of senescent cells. Can the authors quantify the percentage of these variants in all LINE1 elements?

We thank the reviewer for the comments. The 96 upregulated LINE1 variants account for approximately 0.01% of all LINE1 elements. Utilizing L1Base2 for annotation, we found that 1% of these variants are full-length LINE1 with two intact ORFs, and 17.52% are human full-length LINE1 variants exceeding 4500 bp with regulatory potential.

3. More details should be added to the legends. For instances, the abbreviation of RS, SIPS and OIS should be explained in the legends, and this would help readers understand the paper better.

We thank the reviewer for the suggestion. We have included the explanation of RS, SIPS and OIS in the legend of Figure 1 to enhance the manuscript's comprehensibility. The newly added content is as follows: Heatmaps showing the expression of LINE1 subfamilies in cells in the state of stress-induced premature senescence (SIPS), replicative senescence (RS), and oncogene-induced senescence (OIS).

4. Throughout the manuscript, some sentences are hard to understand. Here are some that need to be corrected.

- As in LINE 71, SASP has been explained, there is no need to show the full name of the phrase again in Line 213-214.
- In LINE 250, the reference is missing.
- In LINE 253, the sentence needs to be rephrased.
- In LINE 265, the sentence needs to be rephrased.
- In Figure 1C, the table should be re-organized.
- The spaces are missing before the parentheses that contain the references.

We thank the reviewer for the comments. We have amended the text for clarity on the specified lines and ensured the proper inclusion of references. Additionally, we have reorganized the table in Figure 1C and added spaces before parentheses that contain references.

Referee #3:

I have looked at the manuscript and I indeed have only minor comments, as the other reviewers do, which should/could be corrected and addressed in minor revisions.

We thank the reviewer for the positive comments.

My three 'main' comments (and one very minor, number 4) would be:

1. Most of the work that is performed in the ageing experiments is done with overexpressed proteins. For example, in Figure 2F, the Pax5 ChIP enrichment on LINE 5'UTRs are done with GFP-overexpressed Pax5 fusion. Similarly, the SIRT6 ChIP data in Figure 4F is performed with a GFP-fusion protein overexpressed in HEK293 cells. The 'only' endogenous data the authors present is provided from published data: e.g. encode (H1-ESC cell line for SIRT6) and other not necessarily relevant cell lines (for PAX5). The authors should really attempt to validate, at the very least, the binding of endogenous proteins (PAX5 and/or SIRT6) in their aged cells. Otherwise, the arguments of 'direct' regulation are done in completely artificial setups with regards to the ageing phenotype. Since chromatin regulation is likely to be cell-type specific, I think this is a major point the authors should address.

Figure 2F

Figure 3E

Figure 4F

Figure 4H

We appreciate the reviewer for the suggestion. As the reviewer suggested, we first conducted ChIP assays on both HCA2-hTERT cells that were not irradiated, and senescent HCA2-hTERT cells with anti-PAX5 antibodies, and these results confirmed the binding of endogenous PAX5 to LINE 5'-UTR in the senescence phenotype (Fig. 2F and 3E). Subsequently, we conducted ChIP assays in HCA2-hTERT cells with anti-SIRT6 antibodies. The assays confirmed the binding of endogenous SIRT6 to PAX5 promoter (Fig. 4F). Our previous data confirmed the binding of endogenous SIRT6 to PAX5 promoter in the senescence phenotype (Fig. 4H). We have replaced Fig. 2F, 3E and 4F with the new results and moved the previous data to appendix figure S1, S2A and S3A.

2. Related to their ChIP data, I think key controls are missing. They are easy to address though, and the authors may already have the material: the negative controls for the ChIP are currently done using chromatin of cells that express the empty GFP plasmid. To truly conclude enrichment, the authors should use, in addition, a negative control from the same chromatin (e.g. qPCR analysis of a negative, not-bound target gene of Pax5 and Sirt6 in cells overexpressing the fusion protein Pax5-GFP or Sirt6-GFP), otherwise it is impossible to ascertain if what they see is genomic background.

Appendix Figure S2B

Appendix Figure S3B

Appendix Figure S3C

We thank the reviewer for the suggestion. We have set GAPDH as a negative control of the RT-qPCR in all the ChIP assay we have performed to avoid the genomic background. These results have substantiated the specificity of PAX5 binding to the LINE 5'-UTR, as well as SIRT6's interaction with the PAX5 promoter, further reducing the likelihood that our findings are due to nonspecific genomic background noise. We have included the new data in Appendix Fig. S2B, Appendix Fig. S3B and S3C.

3. The conclusion on Pax5 mediates cellular senescence via LINE1 transcription should be toned down and/or changed. The authors show that inhibiting RT activity in Pax5 overexpressing conditions attenuates molecular (Beta gal) features of senescence, but they do not show that inhibiting LINE1 transcription does so (Figure 3).

We thank the reviewer for the suggestion. Our research has shown that PAX5 directly regulates LINE1 transcription, and that inhibiting retrotransposition activity under PAX5 overexpressing conditions ameliorates senescent phenotypes. These pieces of evidence confirmed that PAX5-activated LINE1 retrotransposition promotes senescence. Previous research has demonstrated that overexpressed LINE1 RNA downregulates the enzymatic function of SUV39H1, leading to heterochromatin loss and the onset of senescent phenotypes in vitro (Valle *et al*, 2022). However, as the reviewer pointed out, whether the PAX5-mediated overexpression of LINE1 RNA directly contributes to the onset of senescence requires further experimentation. Therefore, We have moderated the statements regarding PAX5 mediating cellular senescence through LINE1 transcription in the revised manuscript. The revisions are as follows:

- We have used the term “activating LINE1” rather than “LINE1 transcription” to more accurately reflect how PAX5 mediates cellular senescence through LINE1.
- We have changed the title of the result from “PAX5 promotes cellular senescence by LINE1 transcription” to “PAX5 potentiates LINE1 transcription and promotes cellular senescence”.
- We have changed the caption of Figure3 from “PAX5 promotes cellular senescence dependent on LINE1 transcription activation” to “PAX5 promotes cellular senescence via LINE1 activation”.
- Additionally, we have also added a conclusion at line 239 as follow: “These results suggested that PAX5 boosts LINE1 transcription and promotes cellular senescence.”

4. The authors should specify from early on and on abstract that they are studying human LINE elements.

We thank the reviewer for the suggestion. The abstract has been updated to clearly state that our study pertains to human LINE1 elements.

Dear Dr. Mao,

Thank you for the submission of your revised manuscript. We have now received the enclosed reports from the referees, and I am happy to say that all support its publication now.

Only a few editorial comments will need to be addressed before we can proceed with the official acceptance of your manuscript:

- Please move the conflict of interest statement to after the Acknowledgments section.
- Please remove the author credits from the ms file. All credits need to be entered during ms online submission.
- Please co-submit with your final ms a completed author checklist that can be found here: <https://www.embopress.org/page/journal/14693178/authorguide>. The completed author checklist will also be part of our transparent peer-review process file.
- Table EV1 and its legend need to be removed from the ms file - the table is already provided as a separate PDF file and the legend should be placed in the PDF file. The table is also uploaded as an Excel file with the file format Figure Source Data which could be a mistake? Please only upload one copy of the EV1 table.
- Please address the comments by our data editors:
 1. Please note that the box plots need to be defined in terms of minima, maxima, centre, bounds of box and whiskers, and percentile in the legends of figures 3h, j; 4k.
 2. Please note that information related to n is missing in the legends of figures 3h, j; 4k.
 3. Although 'n' is provided, please describe the nature of entity for 'n' in the legends of figures 2c, f-g; 3a, e-f, i; 4b-c, f, h-k.

I would like to suggest some changes to the abstract that needs to be written in present tense:

As a hallmark of senescent cells, the derepression of Long Interspersed Elements 1 (LINE1) transcription results in accumulated LINE1 cDNA, which triggers the secretion of the senescence-associated secretory phenotype (SASP) and paracrine senescence in a cGAS-STING pathway-dependent manner. However, transcription factors that govern senescence-associated LINE1 reactivation remain ill-defined. Here, we predict several transcription factors to bind to human LINE1 elements and to regulate their transcription by analyzing the conserved binding motifs in the 5'-untranslational regions (UTR) of the commonly upregulated LINE1 elements in different types of senescent cells. Further analysis reveals that the transcription factor PAX5 directly binds to the 5'-UTR of LINE1 and the binding is enhanced in senescent cells. The enrichment of PAX5 at the 5'-UTR promotes cellular senescence and SASP by activating LINE1. We also demonstrate that the longevity gene SIRT6 suppresses PAX5 transcription by directly binding to the PAX5 promoter, and overexpressing PAX5 abrogates the suppressive effect of SIRT6 on stress-dependent cellular senescence. Our work suggests that PAX5 could serve as a potential target for drug development aiming to suppress LINE1 activation and treat senescence-associated diseases.

It is not clear to me whether your study uses human cell lines exclusively? If this is the case, then I think "human" should be added to the title:

The transcription factor PAX5 activates human LINE1 retrotransposons to induce cellular senescence

Regarding the synopsis image, I would like to suggest to remove "prediction" from the upper text in the image. Or change the sentence to "common upregulated transcription factors predicted to bind LINE1".

The second sentence should be changed to "screen with LINE1 5'UTR...".

Given "Young" on the very left, "Senescence" should be changed to "Senescent" on the very right.

Please let me know if you have any questions.

Referee #1:

The authors fully addressed my concerns.

Referee #2:

The authors have now addressed my concerns.

Referee #3:

The authors have addressed all my comments and incorporated all the additional controls.

All editorial and formatting issues were resolved by the authors.

Zhiyong Mao
Tongji university
School of Life Sciences and Technology
1239 Siping Rd
Shanghai 200092
China

Dear Dr. Mao,

I am very pleased to accept your manuscript for publication in the next available issue of EMBO reports. Thank you for your contribution to our journal.

Yours sincerely,
